# Systemic and Intestinal Viral Reservoirs in CD4+ T Cell Subsets in Primary SIV Infection

**DOI:** 10.3390/v13122398

**Published:** 2021-11-30

**Authors:** Xiaolei Wang, Widade Ziani, Ronald S. Veazey, Huanbin Xu

**Affiliations:** Tulane National Primate Research Center, Tulane University School of Medicine, Covington, LA 70433, USA; xwang@tulane.edu (X.W.); wziani@tulane.edu (W.Z.); rveazey@tulane.edu (R.S.V.)

**Keywords:** HIV/SIV, viral reservoirs, CD4+ T cell subsets, primary infection

## Abstract

The HIV reservoir size in target CD4+ T cells during primary infection remains unknown. Here, we sorted peripheral and intestinal CD4+ T cells and quantified the levels of cell-associated SIV RNA and DNA in rhesus macaques within days of SIVmac251 inoculation. As a major target cell of HIV/SIV, CD4+ T cells in both tissues contained a large amount of SIV RNA and DNA at day 8–13 post-SIV infection, in which productive SIV RNA highly correlated with the levels of cell-associated SIV DNA. Memory CD4+ T cells had much higher viral RNA and DNA than naïve subsets, yet memory CD4+ T cells co-expressing CCR5 had no significant reservoir size compared with those that were CCR5-negative in blood and intestine. Collectively, memory CD4+ T cells appear to be the major targets for primary infection, and viral reservoirs are equally distributed in systemic and lymphoid compartments in acutely SIV-infected macaques.

## 1. Introduction

The SIV/macaque model is the premier model used to examine early events in HIV infection. Macaques can be intravenously or mucosally challenged with precisely timed and defined doses of SIV, and tissues can be sampled at necropsy to definitively track the sequences and events in tissues of early SIV transmission and the establishment of infection. Years ago, it was first shown in SIV-infected macaques, and later confirmed in HIV-infected humans, that the intestinal tract plays a fundamental role in the early events of HIV infection. Within days of exposure, regardless of the route of infection, SIV or HIV reaches the intestinal tract, where marked viral replication, amplification, mutation and massive CD4+ T cell depletion occur [1,2,3]. Since the largest populations of activated memory CD4+ T cells reside in the intestinal tract in uninfected hosts, the early peak viral replication that occurs in primary infection is largely attributed to this massive viral replication in this vast pool of susceptible target cells in the gut. It has also been proposed that the productive infection and depletion of intestinal CD4+ T cells are accompanied by intestinal inflammation and the recruitment of additional target cells, supporting even more viral production, resulting in viral dissemination, intestinal CD4+ T cell depletion and the emergence of resistant viral variants that may evade the early immune responses to the transmitting founder viruses [4,5]. To this end, early immune responses likely fail to clear the anatomical viral reservoirs. Thus, the intestinal tract plays a fundamental role in primary and persistent SIV/HIV infection. Preventing the early infection of the intestinal tract in the earliest days of exposure may prevent the establishment of irreversible infection or provide more time for the host to develop more effective immune responses that could clear the infection. In comparison, blood samples can be easily obtained for monitoring viral reservoirs in HIV+ subjects, assessing treatment efficacy and resumption or cure strategies. However, viral reservoirs are also extensively distributed in tissue compartments, yet their size remains unclear in systemic and lymphoid compartments in primary infection. Thus, defining the viral reservoirs in early infected CD4+ T cell subsets and evaluating their size in typical anatomical tissues may help to better understand cellular viral reservoir establishment for HIV cure strategies. However, examining the viral reservoirs in primary HIV infection cannot be performed in humans, so the SIV/macaque model is invaluable for such studies. 

Here, we intravenously inoculated macaques with the highly transmissible and pathogenic SIVmac251, sorted CD4+ T cell subsets from peripheral blood and intestine during primary SIV infection and measured SIV RNA and DNA in the CD4+ T cell subsets. Our data showed that memory CD4+ T cells represented productively infected cells, which contained abundant viral RNA and DNA, with an equivalent distribution of viral reservoir size in systemic and lymphoid compartments in primary infection.

## 2. Materials and Methods

### 2.1. Ethics Statement

Animals were housed at the Tulane National Primate Research Center and studies were conducted in accordance with the recommendations in the *Guide for the Care and Use of Laboratory Animals* of the National Institutes of Health (NIH, AAALAC #000594, IACUC P0305) and the recommendations of the Weatherall report, “The Use of Non-Human Primates in Research”. All clinical procedures were carried out under the direction of a laboratory animal veterinarian. 

### 2.2. Animals and Virus

A total of 8 rhesus macaques (*Macaca mulatta*) from 3 to 16 years of age were used in this study. Eight were intravenously infected with 100 TCID50 SIVmac251 and euthanized for tissue collection at 8 (*n* = 4), 10 (*n* = 1) or 13 days (*n* = 3) after infection. Plasma viral load, CD4+ T cell counts and other data on some of these animals have previously been published [6,7,8]. A chart summarizing the relevant data from the animals in this study is provided in Table 1.

### 2.3. Sorting CD4+ T Cell Subsets

Briefly, peripheral blood mononuclear cells (PBMCs) were isolated from fresh EDTA blood by lymphocyte separation medium (LSM). Intestinal lymphocytes were isolated from jejunum lamina propria cells using EDTA/collagenase digestion and Percoll density gradient centrifugation techniques as previously described [9]. In brief, intestinal segments were incubated in HBSS containing 5 mM EDTA for 30 min, then digested with 60 U/mL collagenase (type II, Sigma, St. Louis, MO, USA) for two consecutive 30 min intervals while harvesting isolated cells. To enrich and purify lymphocytes, isolated cells were layered on a discontinuous 40%/60% Percoll gradient (Sigma, St. Louis, MO, USA), centrifuged for 30 min at 1000 *g*, washed and re-suspended in complete RPMI media containing 5% FCS. Cells from both peripheral blood and jejunum were then enriched using magnetic beads and then 3–4-way sorted with a FACS Aria cell sorter into naïve and memory CD4+ T cells and other subset populations expressing CCR5. All sorted populations were over 95% purity by FACS. 

### 2.4. Genomic DNA and Total RNA Extraction

Fresh PBMC and lymphocytes were processed to extract total genomic DNA and cellular RNA with an AllPrep DNA/RNA Mini Kit (Qiagen, Hilden, Germany) according to the manufacturer’s instructions. Viral RNA in plasma was directly isolated using the QIAamp Viral RNA Mini Kit (Qiagen). The extracted DNA and RNA samples were stored at −80°C until further analysis. 

### 2.5. Quantification of Plasma Viral Load and Cell-Associated SIV RNA and DNA

Plasma viral load and cell-associated SIV RNA/DNA were measured with specific primer sets and probes as previously described [10]. SIV RNA was absolutely quantified by digital droplet PCR (QX100 Droplet Digital qPCR system, Bio-Rad, Hercules, CA, USA). To quantify SIV DNA, a nested PCR was run in parallel to quantify viral DNA, and calibration curves were generated by plotting Cq values against log-transformed concentrations of standard. Copies of SIV RNA/DNA, expressed as copies per 1 million cells, were normalized to cellular input, as determined by copies of genomic CCR5 (single-copy rhesus macaque CCR5 DNA per cell) [11].

### 2.6. Statistical Analysis

Graphic presentation and statistical analysis of the data were performed using GraphPad Prism 9.0 (GraphPad Software, San Diego, CA, USA). Comparisons between groups were analyzed using a paired t test. Values of *p* < 0.05 were considered statistically significant. Correlations between samples were calculated and expressed using the Spearman rank correlation coefficient.

## 3. Results

### 3.1. Levels of SIV RNA/DNA in Peripheral and Intestinal CD4+ T Cells in Primary SIV Infection 

To assess reservoir size in peripheral and intestinal CD4+ T cells in primary SIV infection, we quantified the SIV RNA and viral DNA in sorted CD4+ T cells, which represent SIV transcript and total viral DNA that contains nonintegrated and integrated viral genomes in the SIV life cycle. The results showed that SIV RNA and viral DNA were detectable in CD4+ T cells sorted from the blood and intestinal tissues of acutely SIV-infected animals. In addition, both SIV RNA and DNA maintained considerable levels in target CD4+ T cells, ranging from 8 to 13 dpi, albeit there were significantly higher copies of SIV RNA than SIV DNA in each sample (Figure 1A). There was no significant difference in SIV RNA and SIV DNA levels between blood and intestine-derived CD4+ T cells. Notably, the cellular SIV RNA transcript was highly correlated with the levels of viral DNA in CD4+ T cells (*p* < 0.0001, Figure 1B), which is consistent with the concept that viral DNA could be a template to yield HIV/SIV viral particles in early infection [12,13,14,15]. In general, relative levels of intracellular viral RNA reflect counterpart levels of viral DNA with very few exceptions.

### 3.2. Levels of SIV RNA/DNA in Peripheral and Intestinal Memory CD4+ T Cells during Primary SIV Infection 

It is well known that CD4+ memory T cells are major target cells for HIV/SIV, leading to massive CD4+ memory T-cell infection and subsequent depletion quite early in infection [16,17]. Here, we comparatively analyzed the levels of SIV RNA/DNA in CD4+ memory and naïve cells, which were sorted from blood or intestinal tissues in acute SIV infection. We sorted naïve (CD95neg) and memory (CD95+) CD4+ T cells from both tissues. As shown in Figure 2, there were much higher SIV RNA levels (Figure 2A) and DNA (Figure 2B) in memory CD4+ T cells compared with naïve CD4+ T cells in both blood and jejunum; despite that, naïve CD4+ cells still had considerable levels of SIV RNA/DNA. Notably, there was no significant difference for these viral parameters in naïve or memory CD4+ T cells, regardless of tissue compartment, suggesting that memory CD4+ cells were preferentially infected during primary SIV infection, becoming a major source for virus production and dissemination.

### 3.3. Levels of SIV RNA/DNA in CCR5-Expressing Memory CD4+ T Cells during Primary SIV Infection

With the hypothesis that transmitted founder viruses (TFVs) may depend on CCR5 co-receptor expression of target cells for early infection and dissemination [18,19], we then assessed whether CCR5+ memory CD4+ T cells were initial and dominant target cells in primary SIV infection. The CCR5neg or CCR5+ memory CD4+ T cells were further sorted from these SIVmac-infected animals at the acute stage (Figure 3A). Surprisingly, however, there were no significant differences in SIV RNA or SIV DNA levels between CCR5neg and CCR5+ memory CD4+ T cells sorted from blood or jejunum (Figure 3B,C). From these studies, memory CD4+ T cells in the blood and intestine had significantly and consistently higher levels of both SIV RNA and viral DNA, regardless of CCR5 expression.

## 4. Discussion

Early studies have proven that the CD4 receptor is necessary for viral attachment and infection, so our studies focused on CD4+ T cell subsets. Our data indicated that memory CD4+ T cells are among the earliest cell subsets infected in primary SIV infection, and viral reservoirs in CD4+ T cells are equally distributed in systemic and lymphoid compartments in acutely SIV-infected macaques. In this study, we did not further discriminate between nonintegrated and integrated SIV DNA due to limited numbers of CD4+ T cells from some archived samples. However, nonintegrated viral DNA may be responsible for HIV/SIV production in early infection [12,13,14], and integrated viral DNA serves as the dominant template for persistent HIV replication [15]. Notably, once established in primary HIV/SIV infection, the proviral reservoirs harboring integrated viral DNA are difficult to eradicate by conventional cure strategies [20,21,22]. Thus, viral DNA in CD4+ T cell subsets is a source for viral transcription and virus production, as indicated by the high correlation between viral RNA and viral DNA in peripheral and intestinal CD4+ T cells. With current scalable assays, it is difficult to determine the bona fide replication-competent reservoirs and their distribution in various tissues [23,24,25,26,27,28,29,30,31], e.g., the non-reactivable latency issue with Quantitative Viral Outgrowth Assays (QVOAs) or Tat/rev-Induced Limiting Dilution Assays (TILDAs) and the viral polymorphism problem with near full-length individual proviral sequencing and intact proviral DNA assays (IPDAs). In comparison, qPCR can provide a rapid and sensitive approach to estimate the viral reservoir seeding and size in various tissues, generating comparable data despite overestimating the viral reservoirs due to the existence of defective proviruses. There is no absolute gold standard approach to assess the actual size of viral reservoirs in the body thus far [32].

Early viral reservoirs, once established in primary HIV/SIV infection, are difficult to eradicate by host immune responses or current cure strategies. Firstly, latently infected, quiescent memory CD4 T cells harbor integrated yet transcriptionally silent proviruses in tissues. Secondly, compacted lymphoid tissues such as the germinal center sanctuary structure, shielded from CTL and suboptimal antiretroviral drugs, serve as the major sites for viral reservoirs, leading to the continuous accumulation and/or replenishment of virus-infected cells through clonal expansion, differentiation and migration. Finally, the CTL-dependent MHCI molecule, membrane-anchored CD8 molecule and common γ(c) subunit are downregulated on a fraction of HIV/SIV-infected cells, indicating CTL dysfunction in response to HIV/SIV infection [7,33,34]. Even under antiretroviral therapy, the differential disappearance of inhibitory natural killer cell receptors may impair CTL function [35]. Innate immune responses play an important role in viral containment during primary infection. For example, NK cells are able to efficiently migrate into lymph node follicles and kill the virus-infected cells through HLA-unrestricted cytotoxicity [36,37,38]. However, heterogenous CD3-CD8+ innate immune cells, including NK cells, show delayed proliferation and expansion in early SIV infection [39], accompanied by an elevated proinflammatory “cytokine storm” [40]. Overall, host immune responses are likely compromised to clear HIV/SIV infection during primary infection, predisposing the viral reservoir establishment in early HIV/SIV infection.

Essentially, all intestinal CD4+ T cells in the effector tissues of the gut (diffuse lamina propria) co-express CD69. We reported that all the cells infected in the intestine in very early SIV infection were CD4+CD69+ cells and that the selective loss of CD69+ cells occurred in the blood in very early infection [41]. In addition, the earliest TFV strains of HIV utilize CCR5 as their co-receptor for attachment and entry [42]; we thus compared memory CD4+ T cell subsets expressing CCR5 for viral RNA and DNA levels to those with CCR5 negative. The viral reservoir size in CCR5+ memory CD4+ T cells at the acute stage did not show higher levels than those of CCR5 negative, albeit there were high infection rates in CCR5 negative memory CD4+ T cells.

## 5. Conclusions

In summary, these data indicate that memory CD4+ T cells are the initial and major target cells for early viral replication and dissemination in blood and gut-associated lymphoid tissues, regardless of CCR5 expression. Thus, prevention strategies designed to block the infection of memory CD4+ T cells may prove effective in preventing HIV transmission.

## Figures and Tables

**Figure 1 viruses-13-02398-f001:**
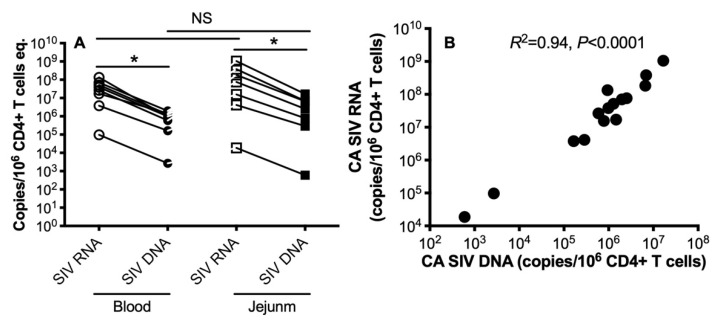
Cell-associated SIV RNA and DNA in peripheral and intestinal CD4+ T cells during primary SIV infection. (**A**) SIV RNA and DNA levels in CD4+ T cells sorted from blood and jejunum at the acute stage. (**B**) Correlation of SIV DNA with SIV DNA in peripheral and intestinal CD4+ T cells at the acute stage. Comparison of SIV RNA/DNA levels per 10e6 cell/equivalents in highly sorted CD4+ T cells as indicated. *, *p* < 0.05. NS, not significant. Note that there is no significant difference between peripheral blood and gut tissue.

**Figure 2 viruses-13-02398-f002:**
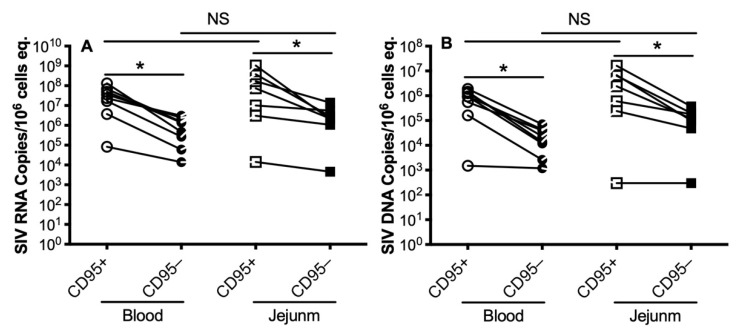
SIV RNA and DNA levels in memory and naïve CD4+ T cells sorted from blood and intestinal tissues during primary SIV infection. Levels of SIV RNA (**A**) and SIV DNA (**B**) in CD4+ T cell subsets sorted from blood and jejunum at the acute stage. Note that significantly higher viral RNA/DNA levels were consistently detected in memory (CD95+) CD4+ T cells compared to naïve (CD95neg) cell subsets. Comparison of SIV RNA/DNA levels per 10e6 cell/equivalents in highly sorted CD4+ T cell subsets as indicated. *, *p* < 0.05. NS, not significant. Note that there is no significant difference between peripheral blood and gut tissue.

**Figure 3 viruses-13-02398-f003:**
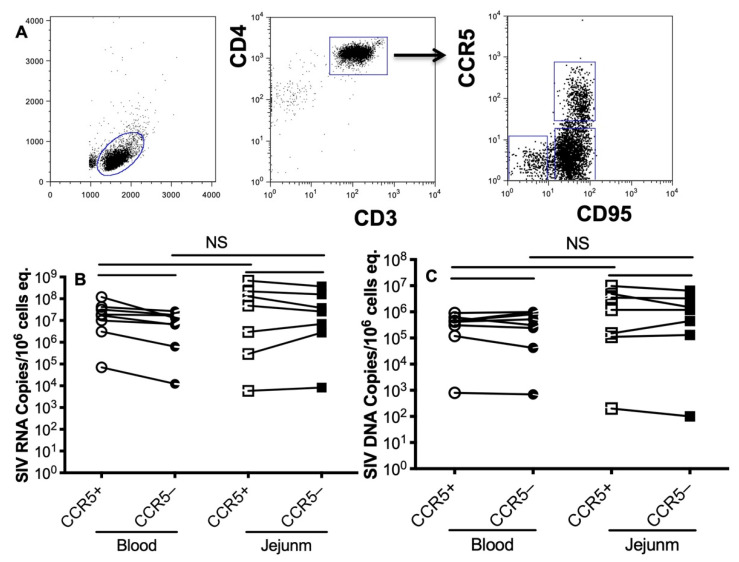
SIV RNA and DNA in CCR5-expressing memory CD4+ T cells sorted from blood and intestinal tissues during primary SIV infection. (**A**) Representative dot plot for cell sorting of CCR5+ or CCR5- memory CD4+ T cells, which was gated on CD3+ lymphocytes. SIV RNA (**B**) and SIV DNA (**C**) levels in CCR5+ or CCR5- memory CD4+ T cell subsets sorted from blood and jejunum at the acute stage. CCR5+ memory CD4+ T cells contain equivalent viral RNA/DNA to CCR5 negative cells in peripheral blood and jejunum. Comparison of SIV RNA/DNA levels per 10e6 cell/equivalents in highly sorted CD95+CCR5+ or CD95+CCR5-CD4+ T cell subsets as indicated. NS, not significant.

**Table 1 viruses-13-02398-t001:** Macaques examined in this study.

Animal	Category	Dose of SIVmac251	Plasma Viral Load (Copies/mL)	CD4+ T Cell Count
T108	8 day p.i.	100 TCID50	57,000	653
BA57	8 day p.i.	100 TCID50	14,000,000	416
HI52	8 day p.i.	100 TCID50	3,700,000	664
HI53	8 day p.i.	100 TCID50	3,600,000	325
AV91	10 day p.i.	100 TCID50	160,000,000	472
M992	13 day p.i.	100 TCID50	35,000,000	227
HI58	13 day p.i.	100 TCID50	8,000,000	651
HI63	13 day p.i.	100 TCID50	24,310,900	438

## Data Availability

Not applicable.

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
