# Peer review of "Systemic and Intestinal Viral Reservoirs in CD4+ T Cell Subsets in Primary SIV Infection"

_viruses, 2021, doi:10.3390/v13122398_

Round 1

Reviewer 1 Report

Wang X et al., examined the SIV DNA and RNA levels in the CD4+ T cell subsets isolated from PBMCs and intestinal tract from the rhesus macaques within 15 days of SIVmac251 infection.  As expected, they found that CD4+ T cells from both tissue sites contained a large amount of SIV RNA and DNA at day 8-13 post SIV infection without ART, where the productive SIV RNA was highly correlated with the levels of cell-associated total SIV DNA. In addition, they found that CD95+ CD4+ T cells had much higher levels of viral RNA and DNA than CD9-CD4+ T cells. However, CD95+CCR5+ CD4+ T cells had no significant difference of reservoir size compared with CD95+CCR5 negative cell populations. They concluded that the memory CD4+ T cells are the major target during the acute SIV infection in both periphery and intestine where viral reservoirs are equally distributed in SIV-infected macaques. This short report provided useful observations to understand the SIV/HIV reservoirs during very early stage of infection. It would be great if the authors can clarify the minor points below:

  1. Lines 38-41 need to be re-structured.
  2. The authors aimed to study the SIV reservoirs in the subsets of the CD4+ T cells in the tissue sites. It is not clear whether effector CD4+ T cells were analyzed with flow cytometry.
  3. In Figures 1-2, it would be great if the integrated SIV DNA was shown in these cells/cell subsets. IPDA might be ideal to better estimate the replication competent SIV since the amount of DNA levels were relatively high.
  4. It is not clear whether CD95 alone is enough to separate memory from naïve CD4+ T cells. Are there any other markers that have been tested to further confirm the memory phenotype of these CD4+ T cell subsets?

Author Response

We are very grateful to reviewer for their time, comments, and opportunity, allowing us to improve our manuscript.  

Q1: Lines 38-41 need to be re-structured.

Response: We thank reviewer for the suggestion. We have revised and polished the description, please see in revised manuscript.

Q2: The authors aimed to study the SIV reservoirs in the subsets of the CD4+ T cells in the tissue sites. It is not clear whether effector CD4+ T cells were analyzed with flow cytometry.

Response: We thank reviewer to point out. We sorted CD95 positive and CD95 negative CD4+ T cells, not combined with CD28 staining for isolation of effector CD4+ T cells. Another reason also lies in a few numbers of effector CD4+ T cells in tissues.

Q3: In Figures 1-2, it would be great if the integrated SIV DNA was shown in these cells/cell subsets. IPDA might be ideal to better estimate the replication competent SIV since the amount of DNA levels were relatively high.

Response: We absolutely agree with your suggestions. As we discussed, we did not further quantify the proviral reservoir size because of some reason: limited numbers of CD4+ T cell sorted from some archived samples; existence of high intact viral genome in SIV+ macaques compared with HIV+ patients even at the chronic stage (Bender AM,et al., 2019) and IPDA not established in our lab.Conceivably, intact SIV genome is overwhelming among the viral reservoirs in primary SIV infection. To address this key point, we are comprehensively assessing the cell-associated SIV DNA/RNA including circular LTR DNA and integrated proviral DNA in systemic and mucosal tissue cells in adult and infant macaques, ranging from 1 to 28 days post SIV infection. Hopefully we could present these data for your concerns.

Q4: It is not clear whether CD95 alone is enough to separate memory from naïve CD4+ T cells. Are there any other markers that have been tested to further confirm the memory phenotype of these CD4+ T cell subsets?

Response: We thank you for your concerns. As we and others reported, rhesus memory/naïve CD4+ T cells could be defined well by consistent CD95 with CD28 surface markers (PMID: 17047153; 19710637; 20795545; 12869504), better than CD45RA, CCR7 and CD45RO (unavailable Ab with cross-reactivity to rhesus) (PMID: 18304631). We feel that CD95 marker is able to define most (if not all) of the memory CD4+ T cells in rhesus macaques.

Reviewer 2 Report

The paper is interesting and well written. I suggest ot improve the paper discussion the role of inhibitory NK cells (see and add as reference paper by Costa et al concerning "differential dissapearance..." published in AIDS).

Author Response

We are very grateful to reviewer for their time, comments, suggestions and opportunity, allowing us to improve our manuscript.  We have added a description and references in the DISCUSSION in relation to host immune responses to HIV/SIV infection. Please see our revised manuscript.